# Phosphate Adsorption from Aqueous Solution Using Electrospun Cellulose Acetate Nanofiber Membrane Modified with Graphene Oxide/Sodium Dodecyl Sulphate

**DOI:** 10.3390/membranes11070546

**Published:** 2021-07-20

**Authors:** Nur Ain Atiqah Mohd Amin, Mohd Akmali Mokhter, Nurrulhidayah Salamun, Wan M. Asyraf Wan Mahmood

**Affiliations:** 1Department of Chemistry, Faculty of Science, Universiti Teknologi Malaysia, Johor Bahru 81310, Johor, Malaysia; ainatiqahkimia@gmail.com (N.A.A.M.A.); nurrulhidayah@utm.my (N.S.); wanmohdasyraf@utm.my (W.M.A.W.M.); 2Advanced Membrane Technology Research Centre (AMTEC), Faculty of Engineering, School of Chemical and Energy Engineering, Universiti Teknologi Malaysia, Johor Bahru 81310, Johor, Malaysia

**Keywords:** electrospinning, cellulose, graphene oxide, adsorption, phosphate

## Abstract

Eutrophication and water pollution caused by a high concentration of phosphate are two concerning issues that affect water quality worldwide. A novel cellulose-based adsorbent, cellulose acetate/graphene oxide/sodium dodecyl sulphate (CA/GO/SDS), was developed for water treatment. A 13% CA solution in a mixture of acetone:dimethylacetamide (2:1) has been electrospun and complexed with a GO/SDS solution. The field emission scanning electron microscope (FESEM) showed that the CA membrane was pure white, while the CA/GO/SDS membrane was not as white as CA and its colour became darker as the GO content increased. The process of phosphate removal from the solutions was found to be aided by the hydroxyl groups on the surface of the CA modified with GO/SDS, as shown by infrared spectroscopy. An optimization condition for the adsorption process was studied by varying pH, immersion time, and the mass of the membrane. The experimental results from phosphate adsorption showed that CA/GO/SDS had an excellent pH adaptability, with an optimum pH of 7, and maximum removal (>87.0%) was observed with a membrane mass of 0.05 g at an initial concentration of 25 mg L^−1^. A kinetic study revealed that 180 min of contact time could adsorb about 87.2% of phosphate onto the CA/GO/SDS membrane. A typical pseudo-second-order kinetic model successfully portrayed the kinetic sorption of phosphate, and the adsorption equilibrium data were well-correlated with the Langmuir adsorption model, suggesting the monolayer coverage of adsorbed molecules.

## 1. Introduction

The growth of urbanization contributes to the rising effect of human activities on aqueous ecosystems [1]. One of the major sources of pollutants is rainwater runoff from urban catchment areas [2]. During a rainstorm, the hazardous pollutants found on the surface of the road, such as heavy metals, fertilizers, and organic contaminants, are easily drawn into rivers and flow into nearby water sources [3]. This occurrence is a significant danger to the climate and to public health [4]. Phosphorus (P) is one of the most common pollutants generated by stormwater run-off [5]. It is also recognized as a valuable resource commonly used in modern agriculture and the chemical industry, and it is usually used as phosphate aqueous solutions [6]. The amount of phosphorus in the water is thought to be a key indicator of eutrophication [7]. When phosphorus becomes the controlling factor, a concentration higher than 0.2 mg/L can cause an algal bloom, which can degrade water quality and cause the death of fish and aquatic plants [8,9]. According to the United States Environmental Protection Agency (USEPA), the allowable limit for the total P concentration of lakes and reservoirs is 0.01 mg L^−1^, in order to decelerate eutrophication [10]. Furthermore, P is a non-renewable resource that will be depleted over the next 100–400 years due to a 3% annual demand growth rate [11]. Hence, phosphate removal and recovery from wastewater is critical for addressing the agricultural and environmental P issue.

Heretofore, some approaches—in particular, biological treatment [12], chemical precipitation [13], electrocoagulation [14], ion exchange [15], and adsorption [16,17]—have been used to remove phosphate from wastewater. Biological and chemical precipitation systems, for example, contain significant volumes of sludge, which can lead to secondary contamination and raise treatment costs [18]. Among these, the adsorption method has been strongly advocated because of its simplicity, low cost, and high removal effectiveness [19]; it is particularly effective at lowering phosphate concentrations to acceptable levels.

Adsorption is a surface process in which contaminants interact with chemical groups on the surface. Porous materials are needed to produce a large specific surface area. It is well understood that adsorbents with a larger specific surface area have a higher adsorption capacity, and the fabrication of porous or fibrous structures would undoubtedly increase their target surface area. The adsorbent, which possesses a large surface area and surface functionality, has a noteworthy effect on the adsorption process. On that account, materials with a sponge-like structure, small particle size, and pollutant-target functional groups are needed to achieve high-efficiency phosphate removal. The evolution of a combined material consisting of all of these features would give a major technical advantage to the water treatment industry. This could be accomplished by electrospinning, a new technique for producing continuous and ultrafine nanofibers [20,21]. Electrospun nanofibers combine the advantages of nanomaterials (large high specific surface area) and bulk materials (easy water separation), which have recently been used as one of the most effective adsorbents [22,23].

Cellulose, which is the most plentiful polymer resource, is a low-cost material. Due to its favorable physical properties, cellulose acetate (CA) is an essential cellulose ester in the industry among the cellulose derivatives [24]. Since it is a readily available, biodegradable, chemically resistant, and inexpensive polymer, it is one of the most extensively used materials in electrospinning [25]. Electrospun CA fibers have a variety of properties, including high porosity and roughness, which make them good candidates for several nanofiltration membrane applications [26]. 

Graphene oxide (GO) is a two-dimensional (2D) form of carbon made up of a hexagonal arrangement of monolayer sp^2^ carbon atoms of oxygenated species, including carbonyl, epoxy, hydroxyl, and carboxylic groups [27]. The negative charge density on the substrate of a basic medium is enhanced by these functional groups in graphene [28]. Nanocomposite fibers focused on GO have attracted a lot of interest due to the fact of their constrained dispersibility, as well as having lots of preferred reactive sites for special chemical modification [29]. GO demonstrates an extremely high surface area, which helps it to have an excellent adsorption capacity. The vast amount of oxygen-containing functional groups and aromatic basal planes within the GO network helps GO to actively attach to several organic and inorganic species via electrostatic interactions, hydrogen bonding, and π-π stacking interactions [30]. As a result, metal ions, dyes, pharmaceuticals, and polycyclic aromatic hydrocarbons have also been studied extensively in the adsorptive removal of contaminants from water using GO [31]. Hence, we combined the properties of cellulose adsorption and the intrinsic properties of graphene oxide to improve the adsorption performance. Previous research has shown that large-scale graphene production using the two-electrode cell system with surfactant assistance produces a very stable graphene suspension [32]. The ionic interactions between the sulphate, nitrate, fluoride, perchlorate, and lithium with graphite are likely to be attributed to the use of a surfactant such as sodium dodecyl sulphate (SDS) in the production of graphene.

The objectives of this work were to prepare a cellulose acetate/graphene oxide/sodium dodecyl sulphate composite (CA/GO/SDS) adsorbent in the form of a membrane and explore the optimal conditions for the adsorption process, such as pH, contact time, and the mass of the membrane. Herein, the CA nanofiber membrane loaded with GO/SDS was prepared via the electrospinning method and the composite membranes were characterized and tested for phosphate ion removal applications. This information will help future studies and other recipients of the novel adsorbent in wastewater treatment.

## 2. Materials and Methods

### 2.1. Materials and Characterization

All chemicals and reagents used for the experiments and analysis were of analytical grade and were used without further purification. All stock solutions were dissolved in distilled water. CA powder was obtained from Sigma–Aldrich (Mn 30,000 g/mol; degree of substitution 39.8%). *N*,*N*-dimethylacetamide (DMac) from Merck and acetone from Sigma-Aldrich were used as solvents. GO/SDS solution was used as received from our prior group. This group has prepared the GO/SDS solution using the two-electrode system [32]. The electrochemical synthesis of graphene was aided by sodium dodecyl sulphate (SDS) as a surfactant, with SDS intercalation to the graphite electrode (anode) followed by the electrochemical exfoliation of the SDS-intercalated graphite electrode (cathode), with an electric current acting as both an oxidizing and reducing agent. The structures and properties of the electrospun nanofiber membrane were characterized by various methods. The binding properties were analyzed using attenuated total reflection-Fourier transform infrared spectroscopy (ATR-FTIR) in the wavelength range of 400–4000 cm^−1^ by placing the sample on the crystal and collecting the data. The surface morphology of the membrane, before and after GO/SDS loading, was identified by field emission scanning electron microscopy (FESEM). The samples for taking FESEM images were coated with gold prior to FESEM examination. The spectrophotometric determination of the phosphate concentration in the solution was carried out by UV/VIS II spectrophotometer nanocolor by measuring the absorbance at a λmax of 690 nm. 

### 2.2. Polymeric Solution and Electrospinning

Fibers with bead-enriched morphology were obtained in the majority of instances. Similar findings have been observed in other places [33,34]. The electrospinning of CA, based on a 13% (*w*/*v*) polymer solution in acetone/DMAc 1:2, resulted in a bead-free nonwoven membrane with polydisperse fiber size in accordance with previous publications [34,35]. For GO/SDS-loaded solutions, GO/SDS was added in 13% by volume concerning the CA content. The spinning solutions were stirred for 1 h to ensure complete dissolution and homogeneity. The fabricated membranes were labelled as CA and CA/GO/SDS. The mixture was loaded into a 10 mL disposable syringe. The polymeric solution was fed at a rate of 2 mL/h through a stainless-steel needle with an inner diameter of 0.643 mm using a syringe pump. A voltage of 20 kV was applied between the needle and the grounded aluminum foil-collector. The length from the needle tip to collector was 15 cm. The relative humidity and temperature in the system were regulated to ensure consistency in electrospinning CA.

### 2.3. Sorption Studies

Potassium dihydrogen orthophosphate (KH_2_PO_4_) was dissolved in distilled water to prepare a stock solution of phosphate (1000 mg L^−1^) and further used for the solution-immersion process by proper dilution. 

The influence of pH on the adsorption of phosphate was investigated by immersing the electrospun CA and CA/GO/SDS nanofiber membranes in 50 mL of phosphate solution at fixed concentrations (25 mg L^−1^) at room temperature. The initial pH values of the phosphate solutions were controlled within the range of 4.0–12.0 with micro-additions of 0.1 M NaOH or 0.1 M HCl. The concentration of phosphate in the residual solution as well as the removal efficiency of the phosphate ions could be determined after reaching equilibrium.

Adsorption studies have been carried out by adding 0.01 to 1.00 g of membrane to a set of beakers filled with 50 mL of phosphate solution at fixed concentrations (25 mg L^−1^) at room temperature. The initial pH of each solution was changed to the optimal value before mixing with the membrane. Samples were withdrawn from the solution and the phosphate concentration was analyzed by using the spectroscopic method based on the blue-colored phosphate of standard test nanocolor ortho-phosphate. A UV-VIS II spectrophotometer nanocolor at 690 nm was used to measure the blue-colored phosphate in the residual solution.

All the adsorption experiments were carried out in triplicate and the differences in the results for the triplicates were addressed. The linear regression method was used to explain the kinetic and isotherm parameters using a Microsoft Excel sheet.

## 3. Results and Discussion

### 3.1. Characterization of CA/GO/SDS Membrane

The findings of FTIR analysis shown in Figure 1 demonstrate the chemical characteristics of the examined materials and the results are shown in the figures below. The GO/SDS spectrum had two main peaks, set at 1639 and 1739 cm^−1^, corresponding to C=C and C=O stretching vibrations, as well as a broad band which extended from nearly 3200 to 3500 cm^−1^, indicating the existence of a hydroxyl group. The absorption peaks represented by the CA spectrum consisted of carbonyl stretching C=O at 1740 cm^−1^, and the hydroxyl group absorption was around 3482 cm^−1^. After being modified with GO/SDS, the intensity of the absorption peaks at 1740, 1368, and 1227 cm^−1^ relative to the acetyl group content decreased. It is also worth noting that the hydroxyl absorption broadened, indicating that the acetyl groups was replaced with hydroxyl groups. 

Figure 2 shows that the digital photos of CA had smooth as well as uniform fibers with diameters ranging from 150 to 300 nm in accordance with previous studies, whereas the CA/GO/SDS had 200–350 nm diameters. The dried membrane had a thickness of 0.0310 cm and this increased to 0.121 cm after the addition of the GO/SDS. Aside from that, the nonwoven surface texture of a single fiber of CA can also be observed in the FESEM micrograph in Figure 2a, and thus validated the choice of the use of the 13% (*w*/*v*) polymer solution in acetone/DMAc 1:2. The FESEM results also confirmed that the incorporation of GO/SDS had an effect on the morphological changes in the electrospun CA membrane through electrospinning. When the GO/SDS solution was added, the smooth CA fibers developed longitudinal surface striations. These striations were thought be caused by a lack of cohesion between the bulk CA and the hydrophilic GO/SDS, which resulted in partial GO/SDS phase separation towards the fiber surface and the formation of loose fiber strands. This was due to the disparity in surface energies of the interacting chemical species [36]. Furthermore, since the GO/SDS diffusivity was much slower than acetone/DMAc, the GO/SDS in the solvent took longer to absorb and adsorbed on the surface, resulting in a striated layer [36]. The electrospun CA fibers’ topographical striated texture was advantageous for the adsorption process. The surface striations from the GO/SDS allowed for more ions to be trapped between the interfaces and the smooth CA fibers [37]. This was intended to ease the permeation of phosphate ions between the membrane and the environment.

### 3.2. Effect of pH

The mechanism of adsorbate uptake can be deducted by understanding the pH effect on the adsorption process. Studies developed by Gonçalves reported that the pH_pzc_ of fresh CA was 5.09 [38]. However, according to this study the results obtained by the pH_pzc_ tests after treatment with the GO/SDS solution indicated that the point of zero charge for CA/GO/SDS changed to 6.2. These changes were expected, since the variation in pH_pzc_ occurred according to the alkalinization power of the GO/SDS solution, resulting in the hydroxylation of the chemical groups of CA.

The point of zero charge (pH_pzc_) theory was used to describe how pH affected ions’ adsorption. The surface charge of adsorbents tends to be neutral if the pH is equal to PZC, and the electrostatic force between ions and the surface of adsorbents can be considered as insignificant [39]. The balance is broken if the pH is larger or smaller than pH_pzc_. When the value of pH is larger than pH_pzc_, the surface of the adsorbents is negative and vice versa. The pH_pzc_ values for the CA and the CA/GO/SDS were found to be 5.3 and 6.2, respectively. Taking this into consideration, the phosphate adsorption was studied on two different membranes, the CA and the CA/GO/SDS, in the pH range of 2.0–12.0. Figure 3 shows that the phosphate removal by the adsorbent increased linearly with pH, reaching an optimum value at pH 6 (CA) and pH 7 (CA/GO/SDS), and then decreased gradually. The poor phosphate adsorption above pH 7 was most likely due to the fact that the higher pH allowed the CA/GO/SDS to bear more negative charge, which was why it favored the repulsion of the negatively charged species in solution. With that in mind, the higher pH values would lower the phosphate uptake, since there was a greater repulsion between the more negatively charged PO_4_^3−^ species and the negatively charged surface sites (OH groups). In the acidic range, the adsorption potential increased with the pH, reaching a maximal removal efficiency of 38.2667% (CA) and 87.0667% (CA/GO/SDS) at pH 6 and pH 7, respectively.

The specific and/or non-specific adsorption of anionic pollutants on the membrane were proposed [40]. The specific adsorption involved ligand exchange reactions (where the OH groups on the surface were replaced by anions) [41]. McGrath et al. discovered that phosphate ions could attach strongly via ligand exchange, outperforming anions that could only attach weakly via both ion exchange and ligand exchange [42]. The non-specific adsorption involved the coulombic force (mostly determined by the pH_pzc_ of the membrane) [40]. The non-specific adsorption illustrated that the phosphate uptake decreased at pH higher than pH_pzc_, corresponding to the negatively or neutrally charged surface of the membrane. Furthermore, the result showed that the maximum phosphate adsorption was attained at the pH 7.0, which was slightly higher than the pH_pzc_ of the CA/GO/SDS (pH_pzc_ of CA/GO/SDS = 6.2). This implies that even though the surface was negatively charged, adsorption still occurred. The phenomenon indicated that the phosphate adsorption by the CA/GO/SDS membrane was primarily a specific process [43].

### 3.3. Kinetic Modelling

Experiments were carried out to determine the kinetics of phosphate adsorption using first-order and second-order kinetic models. In this study, pseudo-first-order and pseudo-second-order were investigated to obtain the rate constants, the equilibrium adsorption capacity, and the adsorption mechanism at different contact times.

#### 3.3.1. First Order Lagergren Model

The first order Lagergren model is presented with the equation given by Lagergren as follows [44]:dq_t_/d_t_ = k_1_(q_e_ − q_t_),(1)
where q_e_ and q_t_ are the amount of phosphate adsorbed on the CA/GO/SDS membrane at equilibrium and at time t (min), respectively, and k_1_ is the rate constant for the pseudo-first-order kinetic model. The amount of phosphate ion adsorbed per unit weight of the membrane was checked at predetermined time intervals. According to the findings, the amount of phosphate adsorption increased with immersion time and reached equilibrium at 180 min. The following time dependent function was obtained by integrating Equation (1) with the boundary conditions t = 0 to >0 (q = 0 to >0):log(q_e_ − q_t_) = log(q_e_) − k_1_t/2.303(2)

The first-order Lagergren model was initially used to interpret the experimental data. The plot of log(q_e_ − q_t_) vs. t would give a linear relationship form in which k_1_ and q_e_ could be resolved by the slope and intercept, respectively (Equation (2)). Figure 4 displayed the kinetics studies of phosphate adsorption onto the CA/GO/SDS membrane. It could be seen that the adsorption process for both the CA and the CA/GO/SDS membrane reached equilibrium at 180 min. The plots were found to be nonlinear, as well as having poor correlation coefficients (Figure 5).

#### 3.3.2. Second-Order Lagergren Model

Lagergren presented the expression of the second-order kinetic model as [45]:Dq_t_/d_t_ = k_2_(q_e_ − q_t_)^2^,(3)
where k_2_ is the rate constant of pseudo-second-order adsorption. The integrated form of the above equation with the boundary condition t = 0 to >0 (q = 0 to >0) is:1/(q_e_ − q_t_) = 1/q_e_ + k_2_t.(4)

The linear form of Equation (4) is:t/q_t_ = 1/k_2_q_e_^2^ + t/q_e_.(5)

The kinetic data were applied to the second-order Lagergren model (Equation (5)). Both q_e_ (cal) and k_2_ were calculated from the slope and intercept of the plot of t/q_t_ versus t, respectively. The plots were found to be linear as well as having excellent correlation coefficients (Figure 6). The theoretical q_e_(cal) values complied well with the experimental q_e_ (exp) values at all interval times studied. This indicated that the second-order model was well-fitted with the experimental data and could be used to agreeably describe the phosphate adsorption on both the CA and the CA/GO/SDS membranes. The pseudo-first-order and the pseudo-second-order models were fitted in the time-dependent adsorption data, and the related kinetics data were compiled in Table 1. As the obtained regression coefficients of the pseudo-second-order model for both the CA and the CA/GO/SDS (*R*^2^ > 0.98) were closer to 1 than the value of the pseudo-first-order model, this phosphate adsorption process possibly tended to be a chemisorption. As a result, a second-order Lagergren model was used to match the experimental results.

### 3.4. Adsorption Isotherm

Adsorption isotherms could favorably explain the mechanism of adsorbates-adsorbents. Based on this study, the commonly used Langmuir and Freundlich isotherm models were applicable to exaggerate the adsorption mechanism of phosphate at different masses of the membranes.

The Langmuir model assumption is that the adsorbent surface is covered by monolayer coverage. The good regression coefficient, *R*^2^, verified that the Langmuir isotherm was a better way to express the equilibrium phosphate adsorption on the CA/GO/SDS membrane at different membrane masses, and this was demonstrated by the standard curve and the calculation formula shown in Appendix A. The agreement of the data of the Langmuir isotherm with the experimental data at all masses of the membrane studied confirmed that the adsorption was monolayer (i.e., assuming that there was no lateral contact with the neighboring adsorbed molecules when a single molecule occupied a single surface site).

The Freundlich model, in contrast to the Langmuir model, allowed for multilayer adsorption on adsorbent [46]. The error functions, however, were higher than the Langmuir isotherm values, proving that the Freundlich model did not match the experimental results. Appendix A lists both the Langmuir and the Freundlich model parameters, as well as the mathematical fits from the adsorption data according to the equation.

### 3.5. Comparison of Phosphate Adsorbents

Table 2 lists the various phosphate adsorbents, the methods used to prepare the adsorbents, the equilibrium times, and their removal efficiencies for comparison. Although the removal efficiency of CA/GO/SDS was not the highest, it was still advantageous because the fabrication of CA/GO/SDS is easy and does not need any post-treatment for cellulose to remove acetyl group via alkali catalyzed saponification [47]. Furthermore, the equilibrium time attained for this new CA/GO/SDS adsorbent was still comparable with that of other adsorbents. The electrospinning process of CA/GO/SDS uses simple equipment which requires less time to produce a nanofiber membrane. 

### 3.6. Method Validation

The analytical performance of CA/GO/SDS based on the adsorption method was investigated and validated: the linearity was performed for different concentrations of PO_4_^3−^ ions in the 1 to 6 mg L^−1^ range using optimum conditions: 25 mL sample volume, pH 7 sample solution, 180 min immersion time, and 0.05 g mass of adsorbent. From Figure 7, it can be seen that the coefficient of determination, *R*^2^, was 0.9922, while in the calculated CA/GO/SDS adsorption method LOD and LOQ were 0.06 mg/L and 0.2 mg/L, respectively. A good repeatability (RSD of 1.66%, n = 3) was obtained.

## 4. Conclusions

A new kind of adsorbent, CA/GO/SDS, in the form of a membrane prepared by electrospinning with the help of acetone and DMAc, looked promising for the removal of phosphate from aqueous solutions. The results of the batch experiments indicated that the highest adsorption capacity of phosphate takes place at a pH of 7.0. Furthermore, the CA/GO/SDS membrane displayed a porous structure and larger specific surface areas to remove the phosphate. The equilibrium adsorption data of phosphate on CA/GO/SDS fitted the Langmuir isotherm model well, with a maximum adsorption capacity recorded as >87.0% with a membrane mass of 0.05 g at an initial concentration of 25 mg L^−1^. The phosphate adsorption kinetics followed a pseudo-second-order kinetic model, implying that the adsorption inclined toward chemisorption. The experimental results corresponded to the Langmuir adsorption isotherm very well, giving the idea of a monolayer coverage of adsorbed molecules. Therefore, the novel CA/GO/SDS membrane could be used as an appropriate adsorbent for phosphate ion removal from water and wastewater.

## Figures and Tables

**Figure 1 membranes-11-00546-f001:**
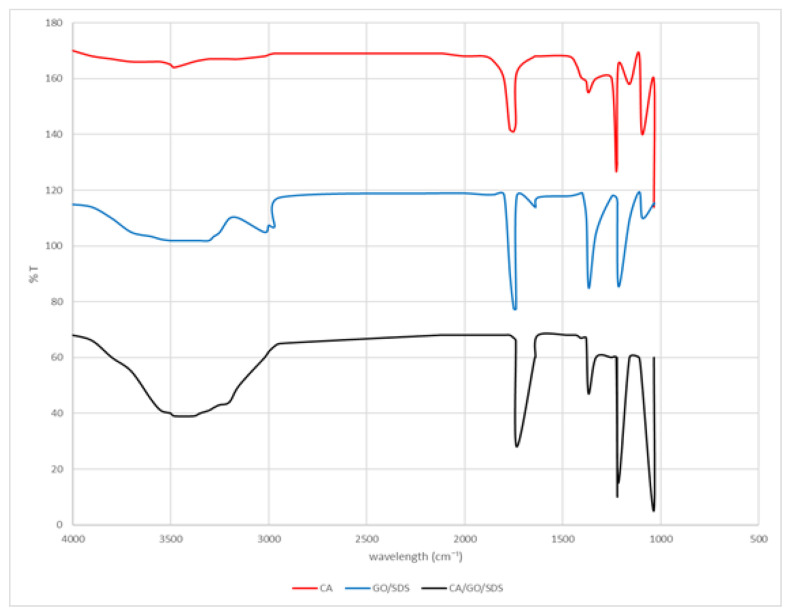
FTIR spectra for CA (red), GO/SDS (blue), and CA/GO/SDS (black).

**Figure 2 membranes-11-00546-f002:**
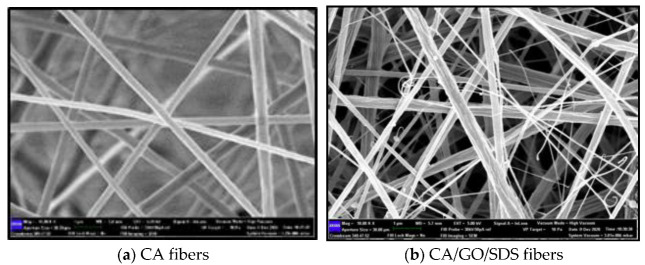
FESEM images for CA and CA/GO/SDS.

**Figure 3 membranes-11-00546-f003:**
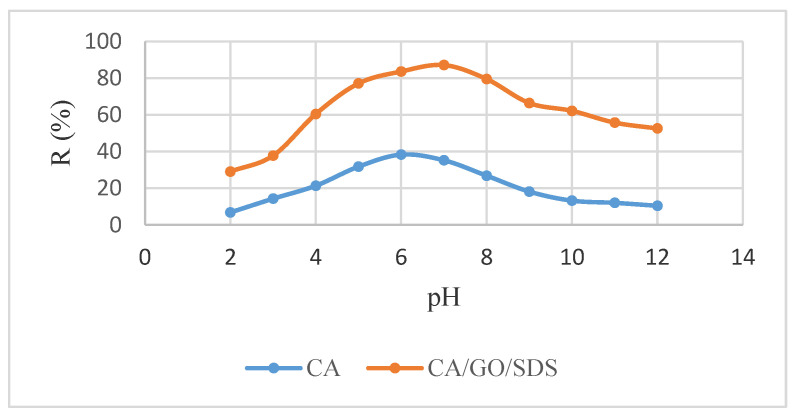
The effect of the initial pH on phosphate adsorption by CA and CA/GO/SDS.

**Figure 4 membranes-11-00546-f004:**
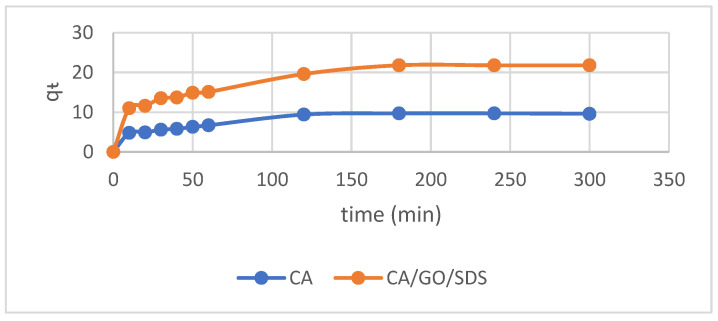
The adsorption time for phosphate removal from the aqueous solution.

**Figure 5 membranes-11-00546-f005:**
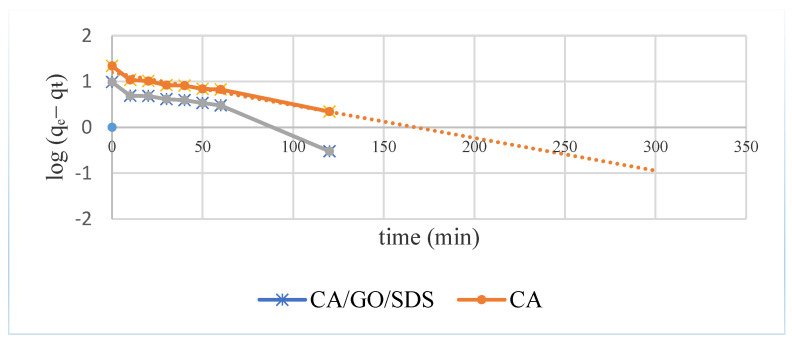
The adsorption kinetics of phosphate adsorption by CA and CA/GO/SDS for the pseudo-first-order model.

**Figure 6 membranes-11-00546-f006:**
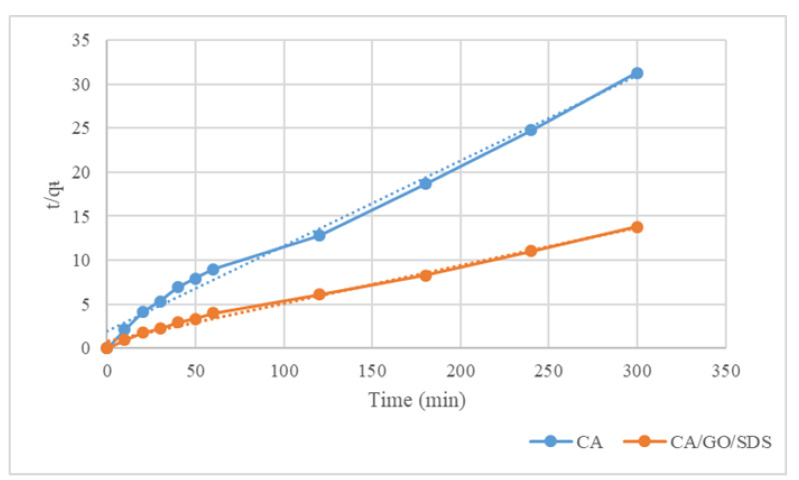
The adsorption kinetics of phosphate adsorption by the CA and the CA/GO/SDS for the pseudo-second-order model.

**Figure 7 membranes-11-00546-f007:**
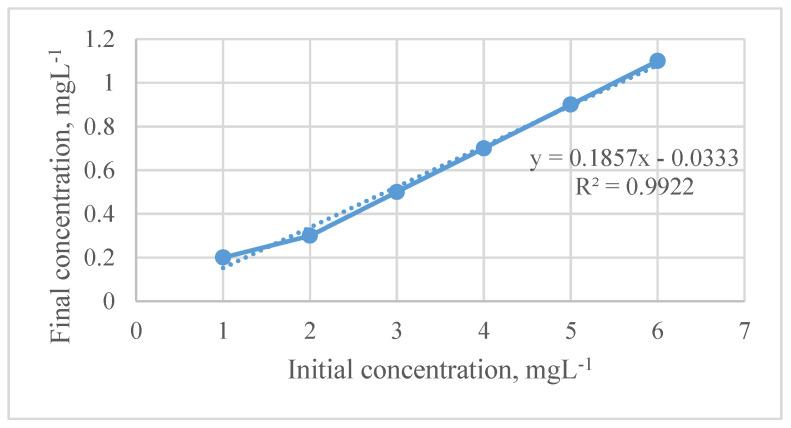
The method calibration curve for PO_4_^3−^ using the CA/GO/SDS membrane.

**Table 1 membranes-11-00546-t001:** Comparison between the experimental and calculated *q*_e_ values for different membranes in the first- and second-order adsorption isotherms at a phosphate concentration of 25 mg L^−1^.

		First Order Adsorption			Second Order Adsorption		
Membrane	*q*_e_ (exp)	*q*_e_ (cal)	*k* _1_	*R* ^2^	*q*_e_ (cal)	*k*_2_ × 10^−3^	*R* ^2^
CA	10.1	9.3994	−0.026	0.9224	10.3627	5.452	0.9896
CA/GO/SDS	21.8	15.7253	−0.0164	0.93	23.2558	2.559	0.992

**Table 2 membranes-11-00546-t002:** Phosphate removal efficiency with various adsorbents.

Adsorbent	Method	Equilibrium Time	Removal Efficiency
Mesoporous zirconia [48]	electrospinning	200 min	92.5%
MgO-functionalized lignin-based bio-charcoal (MFLC) [49]	hydrothermal carbonization and activation	240 min	99.76%
MgO/Fe_2_(MoO_4_)_3_ [50]	blending	80 min	98.38%
La-biochar [51]	pyrolysis and blending	240 min	85%
CA/GO/SDS (present study)	electrospinning	180 min	87.2%

## Data Availability

Not applicable.

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
