# Peer review of "Phosphate Adsorption from Aqueous Solution Using Electrospun Cellulose Acetate Nanofiber Membrane Modified with Graphene Oxide/Sodium Dodecyl Sulphate"

_membranes, 2021, doi:10.3390/membranes11070546_

Round 1

Reviewer 1 Report

Better elaborate the GO/SDS solution described as "received from prior group".

Explain the reason for choosing 13% CA.

Figure 2 misplaced.

Figure 4/5/8 was not referenced anywhere in the manuscript.

Sections 3.3.1 and 3.3.2 needs more data support.

More fundamental explanations required to support the observations.

Need more quantitative information in your conclusions.

Author Response

Dear Reviewers,

Thank you very much for the review of our manuscript and giving us an opportunity to revise the manuscript entitled “Phosphate Adsorption from Aqueous Solution Using Electrospun Cellulose Acetate Nanofiber Membrane Modified with Graphene Oxide/Sodium Dodecyl Sulphate”. We are also grateful to the reviewers for the helpful comments. The manuscript has been revised after carefully considering the comments. 

As described in the following pages, all the comments are addressed clearly and appropriately. In addition, some corrections have been made. All the changes are highlighted in yellow in the manuscript. We believe that the revised manuscript is now acceptable for publication in MDPI.

Kindly refer the document entitled ''Reply to Reviewer 1&2'' attached here. 

Thank you again for your time and consideration of this manuscript. 

Best Regards,

Dr Mohd Akmali bin Mokhter,

Department of Chemistry,

Faculty of Science,

Universiti Teknologi Malaysia

81310 UTM Johor Bahru, Johor, Malaysia

Tel:               Fax :

Reviewer 2 Report

This paper is nothing extraordinary, but there are also no errors.

In the Introduction, there should be a comparison (table?) of other sorbents used for this purpose.

FTIR and SEM analysis do not provide any crucial information. It might as well not be there.

Model fitting is also unnecessary.

In my opinion, this paper should be shortened - delete FTIR, SEM, and models. The authors should focus on the specific properties of adsorbent. The "short communication" will be much better for this results.

Author Response

(The authors gave the same response as above.)

Round 2

Reviewer 1 Report

The authors made sufficient revisions to the manuscript. Moderate English writing revisions are needed for acceptance.

Reviewer 2 Report

I have no more comments. The paper can be published.